# The Not so Good, the Bad and the Ugly: Differential Bacterial Adhesion and Invasion Mediated by *Salmonella* PagN Allelic Variants

**DOI:** 10.3390/microorganisms8040489

**Published:** 2020-03-30

**Authors:** Yanping Wu, Qiaoyun Hu, Ruchika Dehinwal, Alexey V. Rakov, Nicholas Grams, Erin C. Clemens, Jennifer Hofmann, Iruka N. Okeke, Dieter M. Schifferli

**Affiliations:** 1Department of Pathobiology, University of Pennsylvania, School of Veterinary Medicine, Philadelphia, PA 19104, USA; ypwu0902@163.com (Y.W.); druchika@vet.upenn.edu (R.D.); rakovalexey@gmail.com (A.V.R.); gramsn@pennmedicine.upenn.edu (N.G.); 2College of Animal Science and Technology, College of Veterinary Medicine, Zhejiang Agriculture and Forestry University, Hangzhou 311300, China; 3Department of Biology, Haverford College, Haverford, PA 19041, USA; eclemen1@swarthmore.edu (E.C.C.); hofmann.jen@gmail.com (J.H.); iruka.n.okeke@gmail.com (I.N.O.); 4Department of Pharmaceutical Microbiology, Faculty of Pharmacy, University of Ibadan, Ibadan 200284, Oyo State, Nigeria

**Keywords:** *Salmonella*, *S.* Typhi, *S.* Typhimurium, S. diarizonae, PagN, adhesin, invasin, alleles, allelic variants

## Abstract

While advances in genomic sequencing have highlighted significant strain variability between and within *Salmonella* serovars, only a few protein variants have been directly related to evolutionary adaptation for survival, such as host specificity or differential virulence. The current study investigated whether allelic variation of the *Salmonella* adhesin/invasin PagN influences bacterial interaction with their receptors. The *Salmonella enterica, subspecies enterica* serovar Typhi (*S.* Typhi) allelic variant of PagN was found to bind significantly better to different enterocytes as well as to the extracellular matrix protein laminin than did the major *Salmonella enterica, subspecies enterica* serovar Typhimurium (*S.* Typhimurium) allele. The two alleles differed at amino acid residues 49 and 109 in two of the four predicted PagN surface loops, and residue substitution analysis revealed that a glutamic acid at residue 49 increased the adhesive and invasive properties of *S.* Typhi PagN. PagN sequence comparisons from 542 *Salmonella* strains for six representative *S. enterica* serovars and *S. diarizonae* further supported the role of glutamic acid at residues 49 and 109 in optimizing adhesion to cells and laminin, as well as for cell invasion. In summary, this study characterized unique residues in allelic variants of a virulence factor that participates in the colonization and invasive properties of different *Salmonella* stains, subspecies and serovars.

## 1. Introduction

*Salmonella enterica* subsp. *enterica* (*S. enterica*) is an entero-invasive bacterial pathogen that utilizes a type three secretion system (T3SS) encoded on the *Salmonella* pathogenicity island 1 (SPI-1) to invade intestinal epithelial cells. T3SS-driven uptake is particularly critical for intestinal *S. enterica* infections in mammals, as demonstrated with several experimental animal models [1]. *S. enterica* strains are differentiated by their flagella and O-antigens, which are highly variable and provide the basis for identification of over 2500 different serovars [2,3]. There is a direct correlation between the range of host adaptation of different *S. enterica* serovars with their levels of virulence and types of pathogenesis [4]. Serovars that are better adapted to a specific host species, such as the human-restricted *S. enterica* serovar Typhi (*S.* Typhi), are extremely pathogenic due to their ability to leave the intestines and spread hematogenously, resulting in sepsis. In contrast, serovars such as *S. enterica* serovar Typhimurium (*S.* Typhimurium) that have a broad host range are restricted to local invasion, and their containment by host inflammatory responses narrows their pathogenesis to gastrointestinal symptoms in humans. Finally, some *Salmonella* such as *S. enterica* subsp. *diarizonae* (*S. diarizonae*) are primarily associated with cold-blooded animals, and only rarely result in invasive diseases in sheep or humans [5,6].

Even though all *Salmonella* have and express SPI-1 genes [7], variations in the sequence of specific SPI-1 proteins and the unique repertoire of translocated SPI-1 effector proteins by different species, serovars and even strains within the same serovar has a direct impact on the invasion efficiency of different cell types [8,9,10]. Moreover, the diversity of *S. enterica* invasion levels for various cell types and host species can be partially attributed to variations in the regulation and export efficiencies of effector proteins through the SPI-1 T3SS [11]. In addition to the SPI-1 T3SS, cell invasion by *S. enterica* involves two outer membrane proteins (OMP), Rck and PagN, that have been designated adhesins–invasins due to their ability to promote both bacterial binding to host receptors and cellular uptake [12,13,14]. Rck has several adhesive properties: it binds to factor H to mediate bacterial resistance to complement and adheres to laminin and interacts with the epidermal growth factor receptor for a zipper mechanism of cell invasion [13,15,16,17]. Rck also contains a self-association motif that has the potential to mediate interbacterial attachment, as described for Hra1/Hek, an integral outer membrane protein of enteroaggregative *Escherichia coli* (*E. coli*) [18,19]. The adhesins/invasins Hra1/Hek, Hra2 and Tia of enteroaggregative or enterotoxigenic *E. coli* share homologies with membrane spanning domains of PagN, but their surface-exposed loops are less similar, alluding to variable binding affinities for different host receptors [18,20,21,22].

Unlike *rck,* which is plasmid-encoded and absent in many virulent *S. enterica* serovars and strains, *pagN* is encoded on the bacterial chromosome and present in most, if not all, *S. enterica*. One study indicated that deletion of *pagN* in *S.* Typhimurium SL1344 grown under PhoPQ-activating conditions to inhibit SPI-1 gene transcription decreased bacterial adhesion and invasion of HT-29 cells [12]. In contrast, deletion of *pagN* in the *S.* Typhimurium strain LT2 did not impact adhesion to HT-29 or other cells [23], potentially due to the expression of other adhesins such as the type 1 fimbriae induced by the growth culture conditions. More surprisingly, despite the presence of both SPI-1 T3SS and type IV pili that can each independently mediate invasion, deletion of *pagN* impacted both adhesion and invasion of HT-29 cells in a strain of *S.* Typhi [23,24]. Although the culture conditions used might not have activated functionally detectable T3SS expression, type IV pili were still functional, since a *S.* Typhi mutant lacking type IV pili affected invasion as much as the ∆*pagN* mutant [23]. The latter result suggested that both PagN and the type IV pili contribute to *S.* Typhi invasion of cells. In addition, oral challenge of iron-overloaded Swiss Albino mice with wild type and *pagN* mutants of *S.* Typhi indicated that the former strain was more invasive and had increased lethality at an infectious dose of 2 × 10^7^ CFU and in competition assays. Finally, even though no differences were observed between the wild type and *pagN* mutant strains of *S.* Typhimurium [23], use of PagN from each serovar as an immunogen demonstrated some level of immune protection in mice challenged with the wild type strain from the corresponding serovar [23]. Thus, the role of PagN in the invasive properties of *S.* Typhi and *S.* Typhimurium remain somewhat controversial and studies of their function are likely impacted by the confounding effects of additional adhesins/invasins, different levels of PagN expression, and/or variations in the protein sequence of PagN.

Based on these results and our previous findings on FimH adhesin alleles in various *S. enterica* serovars and strains for host cell binding specificities [25,26], we wondered whether allelic variation in PagN [10] modulates bacterial adhesion and invasion. Here we determined that the *S.* Typhi allelic variant of PagN provides significantly better bacterial binding and invasive efficiencies for different enterocytes and the extracellular matrix protein laminin, as compared to the major *S.* Typhimurium allele. To our surprise, PagN from *S. enterica* subsp. *diarizonae* was also significantly more adhesive and invasive than the major *S.* Typhimurium allele. Sequence comparison and functional analysis of substitutions as specific residue positions revealed that amino acids in two of the four PagN surface-exposed loops contributed to these phenotypes. These results further support the role of allelic variation of virulence factors in adjusting the level of pathogenic attributes among *Salmonella* subspecies and serovars.

## 2. Materials and Methods

### 2.1. Bacterial Strain and Plasmid Constructions

The bacterial strains and plasmids used in this study are described in Table 1 and all PCR primers are listed in Appendix A. Unless stated otherwise, all the reagents were from MilliporeSigma (St. Louis, MO, USA). Bacteria were routinely grown in LB-Lennox media unless otherwise indicated. When appropriate, ampicillin (200 µg/mL) or kanamycin (45 µg/mL) was added to the growth medium. For plasmid constructions, the *pagN* genes were amplified from *S.* Typhimurium, *S.* Typhi and *S. diarizonae* genomic DNA by PCR with the Q5 High-Fidelity DNA Polymerase (New England Biolabs Inc., Ipswich, MA, USA). For PagN expression the allelic genes were cloned into AHT-inducible plasmid pRS1 using Gibson assembly with amplicon prepared from appropriate primers. Three site-directed substitution mutants were prepared by Gibson assembly of pRS1 and amplicons prepared with *pagN* external and internal primers for *S.* Typhimurium or *S.* Typhi, or amplicons prepared with mutagenic primers. A *pagN-his* fusion construct was prepared by insertion of a *Nde*I-*Hind*III restricted amplicon into pET22b. *S.* Typhimurium SL1344 *sipB::aphA-3* mutant was prepared by P22 generalized transduction from strain DMS1507 [27,28] and SL1344 *sipB::aphA-3* ∆pagN was constructed by Gibson assembly and allelic exchange as described [29,30,31]. All plasmid and strain constructs were confirmed by PCR and sequencing.

### 2.2. Protein Expression and Antibody Preparation

The histidine-tagged PagN was expressed from the pET22b construct using the IPTG inducer and isolated by metal chelation chromatography as described previously [35]. A specific polyclonal antiserum against PagN-His was prepared in rabbits by using a conventional immunization protocol (Cocalico Biologicals Inc., Reamstown, PA, USA). The antiserum was adsorbed with *E. coli* BL21(DE3)/pET22b before use as described previously [36]. Briefly, 1 mL antiserum with 0.06% sodium azide was incubated with bacterial pellets from 10-mL cultures grown overnight for 18 h at 4 °C. After three adsorption cycles, the antiserum was filtered (0.02 µm-pore-size) before use. PagN expression in *E. coli* AAEC189 for all the binding and invasion studies was induced with AHT by growing the bacteria overnight at 30 °C, or for 2 h at 37 °C, starting with log phase cultures (A_600_ = 0.3). Comparable levels and bacterial surface expression of the three cloned PagN alleles were standardized by using various concentrations of inducers (0.005–2 µg/mL AHT) for Western blot analysis and ELISA, as done previously [25]. Outer membrane proteins were prepared as described previously [37].

### 2.3. Cell Cultures

The human colonic cell line RKO (ATCC CRL2577) was cultured in Dulbecco’s Modified Eagle Medium (DMEM; Invitrogen, Life Technologies) supplemented with 15% (*v*/*v*) heat inactivated fetal bovine serum (FBS) and antibiotics to a final concentration of 100 U/mL penicillin and 100 µg/mL streptomycin (Gibco, Life Technologies) [25]. The porcine cell line IPEC-J2 (DSMZ ACC 701) was cultured with 15% heat inactivated FBS (Sigma-Aldrich), 1% penicillin/streptomycin, 1% insulin/transferrin/selenium (Gibco) and 5 ng/mL epidermal growth factor (Sigma) in DMEM/F-12/HAM (1/1/1, *v*/*v*/*v*; Gibco). The cells were incubated at 37 °C in a humid atmosphere with 5% CO_2_.

### 2.4. Bacterial Binding and Invasion Assays

Both epithelial cell cultures grown to confluence in 24-well plates (Corning, CLS3596) were used for the binding assays with recombinant *E. coli* AAEC189 (*E. coli* ∆*fim*) with pRS1 plasmid constructs s expressing a *Salmonella* PagN allele. Bacteria were grown overnight, diluted 10^−2^ in LB broth with inducers (see above), incubated for 16–17 h, washed three times with PBS and diluted in DMEM to inoculate with a multiplicity of infection of 100 bacteria (in 0.25 mL) to 1 enterocyte. Exact bacterial inoculum numbers were checked by standard CFU counts. The culture plates were centrifuged (600 *g*, 5 min) to initiate contact between the bacteria and the cells and incubated for 1 h at 37 °C in 5% CO_2_. To evaluate cell adhesion, the infected monolayers were washed thrice with PBS to remove non-associated bacteria and treated with 0.5% Triton X-100 to release and count the cell-associated bacteria by CFU enumeration. For the invasion assays, a standard gentamicin protection assay was performed [38]. Following a 1 h infection (see above), cells were incubated with medium containing gentamicin (100 µg/mL) for 90 min at 37 °C in 5% CO_2_. After three washes with PBS, bacteria were released with Triton X-100 and enumerated as described above. All experiments were done in triplicate wells and repeated at least thrice.

### 2.5. Microscopy

For fluorescence microscopy, *E. coli* strain AAEC189 carrying plasmids pRS1, pDMS1973 (*pagN*_Ty_), pDMS1974 (*pagN*_Tm_) or pDMS2062 (*pagN*_di_) were grown and induced for PagN expression as described above. The bacterial cells were deposited on slides, dried for 20 min and washed once with PBS. Bacteria were fixed with 4% paraformaldehyde in PBS, pH 7.4, then labeled with adsorbed anti-PagN antisera (1:500), followed by anti-rabbit Alexa Fluor 480 (1:1000; Invitrogen, Life Technologies, Grand Island, NY, USA). Images were captured with a Coolsnap digital camera (Photometrics, Tucson, AZ, USA) mounted onto a Nikon Eclipse E600 microscope with Coolsnap version 1.2.0 software (Roper Scientific, Tucson, Arizona).

### 2.6. Binding to Extracellular Matrix Proteins

Immuno Maxisorb plates with 96 wells (Nunc; Thermo Fisher Scientific, Rochester, NY, USA) were coated with 10 μg/mL of human collagen I, chicken collagen II, human collagen IV, bovine fibronectin, murine laminin or BSA (Sigma-Aldrich) in PBS at 4 °C overnight and then washed with PBS and blocked with PBS plus 1% BSA for 2 h. Binding of bacteria to laminin coated on plates was studied by using PagN-expressing *E. coli*. Bacteria grown and induced to make PagN were centrifuged, suspended in PBS to 10^7^ CFU in 100 µL and added to laminin coated wells. Bacteria harboring empty plasmid pRS1 were used as a control. After incubation for 1 h at 37 °C, unbound bacteria were removed by three washing cycles, anti-PagN antiserum (1:500) was added, followed by wash cycles and incubation with goat anti-rabbit HRP-conjugated antibody (1:2000). After three wash cycles, bound antibodies were detected by using the 1-Step Turbo TMB ELISA substrate (Thermo Fisher Scientific) followed by 2 M sulfuric acid and measuring the absorbance at 450 nm. For the binding inhibition assays, double dilutions of heparin or heparan sulfate (400–0.4 µg/mL) were incubated with bacteria for 1 h and the mixtures were added to the laminin-coated wells for further processing as described above.

### 2.7. Bacterial Genomes and PagN Sequences

*Salmonella* genomic sequences from 497 previously studied *S. enterica* from 6 serovars with two biovars (serovars Typhi, Dublin, Choleraesuis, Typhimurium, Enteritidis, Newport and Gallinarum, with biovars Gallinarum and Pullorum for the latter serovar) and from 45 confirmed *S. diarizonae* (Appendix A) were obtained from NCBI RefSeq database or assembled from NCBI SRA, EBI ENA and Wellcome Sanger Institute repositories as described [10]. The genomes were used to determine their encoded PagN sequences. The genomes of 13 incorrectly serotyped *S. diarizonae* strains were detected with SISTR (their corrected serovar attribution was added to Appendix A) and not analyzed for PagN [39]. Protein alignments were done with Megalign, DNASTAR Lasergene (Madison, WI, USA).

### 2.8. PagN Structure Analysis

The 3D-structural model of PagN was predicted by using the corresponding sequence from *S.* Typhimurium LT2 and I-TASSER [40,41]. Among the five best predicted models, the 3rd model was chosen to be shown in Figure 1B (Protean 3D, DNASTAR Lasergene), with an overall ERRAT quality factor of 84.4 [42], in agreement with a published model for *S.* Typhi PagN [43]. The five best I-TASSER predicted *S. arizonae* PagN structures essentially overlapped with the ones of *S.* Typhimurium and *S.* Typhi.

### 2.9. Statistical Analysis

Student’s non-paired *t* test (two tailed) was used with Prism 8 (GraphPad Software, San Diego, CA, USA) to calculate statistical significance for all the binding assays (* *p* < 0.05, ** *p* < 0.01, *** *p* < 0.001 and **** *p* < 0.0001).

## 3. Results and Discussion

PagN of *S.* Typhimurium and *S.* Typhi had similar sequences with two to three substitutions in predicted surface-exposed loops 1 and 2 (Figure 1). Since PagN of both strains act as adhesins and invasins for human intestinal epithelial cells, we wondered whether these allelic variants differentially impacted these properties. To ensure consistent and comparable expression of PagN, we cloned each gene into inducible expression plasmids to make pDMS1973 and pDMS1974 for the expression of PagN from *S.*Typhi (*pagN*_Ty_) and *S.* Typhimurium (*pagN*_Tm_) respectively. Western blot analysis of outer membrane preparations from *E. coli* AAEC89 carrying one of these two plasmid constructs detected bands specific for each PagN allele (Figure 2A).

Moreover, both alleles of PagN were detectable on the bacterial surface by immunofluorescence (Figure 2B, panel a to f) and ELISA (not shown). Not surprisingly, the strongest signal was observed for the *S.* Typhimurium PagN allele, as this was the immunogen used to prepare the antiserum.

The *E. coli* strain AAEC189 lacks adhesive type 1 fimbriae, providing us with a bacterial context in which to study PagN-mediated bacterial binding and cellular uptake free of known *Salmonella* adhesins and invasins. Using both RKO and IPEC-J2 enterocytes, we found that both PagN_Ty_ and PagN_Tm_ mediated bacterial binding and invasion, as previously reported for ovarian hamster epithelial-like CHO-K1 cells and human colonic HT-29 cells [12,23]. More importantly, comparisons of the two alleles indicated that PagN_Ty_ was more efficient than PagN_Tm_ for both cell adhesion and invasion (Figure 3A), supporting an allelic variant effect on a virulence property. Additional studies with a *S.* Typhimurium *pagN* and *sipB* (essential SPI-1 translocon subunit) deletion mutant complemented with different PagN-expressing plasmids did not show significant different levels of adhesion/invasion of RKO cells (data not shown), suggesting a dominant phenotype due to the expression of one or more other *Salmonella* adhesin(s) and invasin(s) under the used growth conditions. Thus, the use of *E. coli* with a controlled expression system allowed us to bypass the masking effect of additional *Salmonella* adhesins/invasins differentially expressed in different serovars and environmental growth conditions and identify functional differences due only to sequence variations of PagN alleles [44,45].

PagN sequence alignments for 52–75 strains for each of the seven *S. enterica* serovars, including *S.* Typhimurium [10], highlighted that all PagN had aspartic acid residues at position 49 and 109, with the exceptions of *S.* Typhi that had a glutamic acid at position 49, and a group of *S.* Newport strains that had a glutamine at position 109 (Figure 1A). Since the allelic PagN proteins of all strains of *S.* Typhi and *S.* Typhimurium varied at these two positions and were each predicted to be located in a different bacterial surface loop (Figure 1B), we investigated their relative involvement in bacterial adhesion and invasion. For this, we generated PagN_Tm_ with a D49E or a D109Q substitution and expressed them in *E. coli* strain AAEC189. Although both mutated PagN mediated better bacterial binding to RKO cells than did PagN_Tm_, only the D49E substitution in PagN had a significantly stronger effect on bacterial invasion (Figure 3B). This result highlighted the contribution of the *S.* Typhi glutamic acid at position 49 for the improved interaction of PagN with a host cell receptor to promote bacterial uptake. This result is consistent with a role for the PagN allele in the increased pathogenicity of *S.* Typhi relative to *S.* Typhimurium pathogenesis following human infection, in agreement with the former serovar’s hematogenous bacterial spreading in humans resulting frequently in sepsis, in contrast to the latter serovar and its pathogenesis that is usually contained in the gastro-intestinal organs.

Other than subsp. *enterica,* most subsp. of *S. enterica* are associated with cold-blooded animals. Therefore, we expected that a construct expressing the PagN allele of these non-*enterica* subspecies would neither bind nor efficiently invade human RKO cells and therefore would serve as a negative control. Thus, we cloned the *S. diarizonae* PagN (PagN_di_) in the same inducible expression plasmid used for the two other PagN alleles to make pDMS2062 and confirmed protein production (Figure 2A) and surface expression (Figure 2B, panels g and h). To our surprise, *E. coli* making PagN_di_ were significantly more adhesive and invasive than the ones expressing PagN_Tm_ (Figure 4A).

PagN alignments of 48 available strains of *S. diarizonae* highlighted that they all had an aspartic acid at position 49, like most evaluated subsp. *enterica* serovars (Typhi being the exception, Figure 1A). In contrast, unlike the subsp. *enterica* serovars, all PagN_di_ carried a glutamic acid at position 109, suggesting that a D109E substitution at this position would be functionally significant. In support, a D109E substitution in the *S.* Typhimurium PagN (pDMS 2083) improved not only bacterial binding, but also bacterial uptake by RKO cells (Figure 4B), indicating that glutamic acid in this position optimizes the function of PagN as both an adhesin and invasin. Whether other substitutions in PagN_di_, such as the arginine or lysine at position 150 and 154 in loop 3 (Figure 1) modulate bacterial invasion remains to be determined. Like *S.* Typhi, *S. diarizonae* have type IVB pili involved in intestinal cell invasion, raising the possibility that the concomitant increased binding property of the *S. diarizonae* PagN allele contributes to the reported increased bacterial virulence in humans under specific conditions [5]. Taken together, the results showed that PagN alleles with a glutamic acid residue at either 49 or 109 improves bacterial binding and/or invasiveness, possibly due to the more ionizable and long chain characteristics of glutamic acid relative to aspartic acid and glutamine.

In addition to promoting bacterial attachment to cells, many bacterial adhesins interact with various host glycoprotein of the extracellular matrix, as exemplified by the ability of PagN_Ty_ to bind to laminin [23,46,47,48]. *S. enterica* may encounter laminin either on intestinal areas denuded of epithelial cells (e.g., extrusion zones at villi tips) or in the subepithelial space after invading enterocytes and crossing the intestinal epithelial layer. Notably, the low pH and magnesium concentrations in *Salmonella*-containing vacuoles (SCV) of invaded cells induce the PhoPQ two-component system responsible for PagN expression [49,50,51]. Thus, whereas the *S. enterica* SPI-1 system is induced in the intestinal environment and used for enterocyte invasion, expression of PagN in SCV (where SPI-1 is largely repressed) may prepare *Salmonella* for future cycles of cell invasion after escape from intestinal epithelial cells. Our side-by side comparison of PagN_Ty_-, PagN_Tm_- and PagN_di_-expressing *E. coli* showed that all three bacteria adhered significantly to laminin, albeit the former better than the latter two (Figure 5A). None of the bacteria bound to collagen I, II or IV or fibronectin (not shown).

To determine the potential role of PagN residue 49 and 109 in PagN_Ty_ binding to laminin, we tested the three mutants described above. As for the intestinal epithelial cell binding results, bacteria with a glutamic acid at position 49 of PagN_Tm_ (D49E) bound significantly better to laminin than did bacteria with PagN_Tm_ (Figure 5B). Glutamic acid or glutamine at position 109 of PagN_Tm_ (D109E and D109Q) also increased binding, albeit less efficiently.

Since both PagN and heparin bind to laminin [52], we next determined whether heparin or heparan sulfate could inhibit PagN-mediated bacterial binding to laminin. In contrast to the reported inhibitory effect of heparin on PagN-mediated invasion of CHO-K1 cells [53], neither heparin nor heparan sulfate at concentrations as high as 400 µg/mL interfered with the binding of PagN_Ty_-expressing bacteria to laminin (data not shown). These combined results suggested that PagN and heparin bind to different laminin sites and that the invasion of CHO-K1 cells by PagN-expressing bacteria is laminin-independent. Thus, our studies highlight the independent adhesive properties of PagN in binding to either intestinal cells or the extracellular matrix protein laminin, two relevant targets for *Salmonella* host invasion.

In summary and together with our previous studies on variants of fimbrial adhesins [25,26,54], this study on PagN, a *Salmonella* adhesin/invasin, further supports the importance of protein sequence allelic variants in virulence properties [55], including pathogenic properties such as adhesion and invasion [4].

## Figures and Tables

**Figure 1 microorganisms-08-00489-f001:**
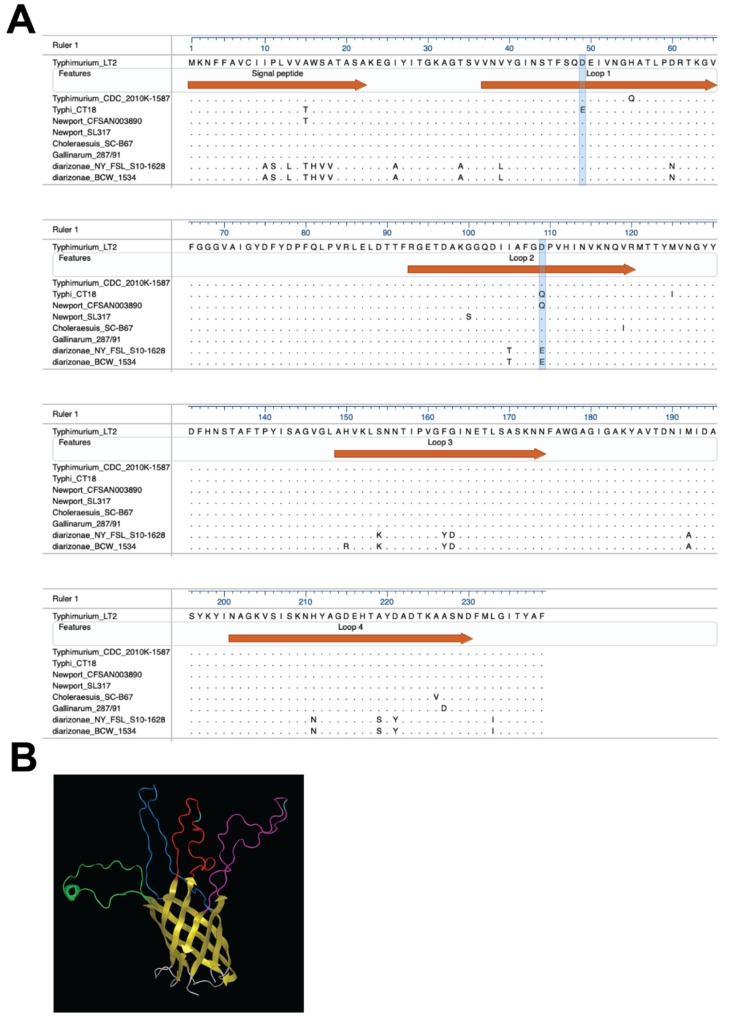
PagN sequences and predicted structure. (**A**) Alignment of PagN alleles from different *Salmonella* strains (Appendix A) [10] each representing N strains from one or more serovars, or clusters within serovars or subspecies with the same sequence in the surface loops of mature PagN; *S.* Typhimurium_LT2 (*n* = 68) with same sequence for *S.* Enteritidis (*n* = 74), *S.* Pullorum (*n* = 29), *S.* Dublin (*n* = 74), and a cluster of Newport strains (*n* = 25); *S.* Typhimurium_CDC_2010k-1587 (*n* = 7); *S.* Typhi_CT18 (*n* = 75); *S.* Newport_CFSAN003890 (*n* = 23); *S.* Newport_SL317 (*n* = 22); *S.* Choleraesuis_SC-B67(*n* = 74); *S.* Gallinarum 287/91(*n* = 23); *S. diarizonae* NY_FSL_S10-1628 (*n* = 3) and *S. diarizonae_*BCW_1534 (*n* = 45). (**B**) *S.* Typhimurium mature PagN model predicted with I-TASSER. Color codes are for the membrane spanning beta-barrel (yellow), the periplasmic loops (grey), surface-exposed loop 1 (purple) with residue 49 (turquoise), loop 2 (red) with residue 109 (turquoise), loop 3 (green) and loop 4 (dark blue).

**Figure 2 microorganisms-08-00489-f002:**
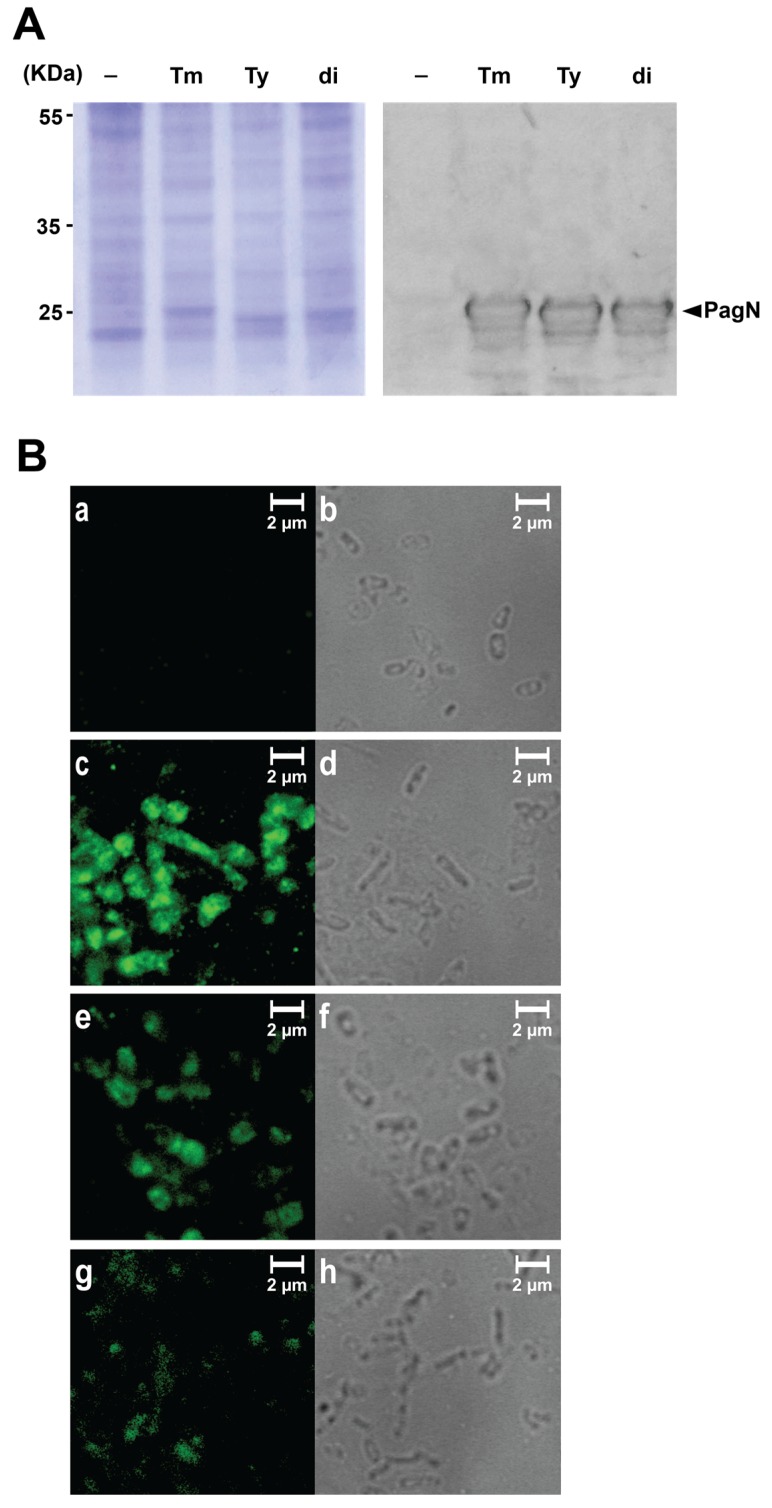
Expression of pagN alleles from *S.* Typhimurium, *S.* Typhi and *S. diarizonae*. (**A**) Expression of PagN in *E. coli* AAEC189 transformed with pRS1 (empty vector, -), pDMS1973 (*pagN*_Ty_), pDMS1974 (*pagN*_Tm_) or pDMS2062 (*pagN*_di_) was induced by AHT (0.2–0.4 µg/mL) for 2 h at 37 °C. Isolated outer membrane proteins analyzed by SDS-PAGE, followed by Coomassie blue staining or western blotting with anti-PagN antisera showed a clear band for PagN expression at 25 kDa. (**B**) Visualization of PagN surface expression in *E. coli* AAEC189 expressing the allele from *S.* Typhimurium (**c**,**d**), *S.* Typhi (**e**,**f**), *S. diarizonae* (**g**,**h**) or empty vector as negative control (**a**,**b**). Phase-contrast microscopy (right) and fluorescence microscopy (left) were used to detect bacteria labeled with anti-PagN antisera, followed by Alexa Fluor 488-conjugated anti-rabbit IgG.

**Figure 3 microorganisms-08-00489-f003:**
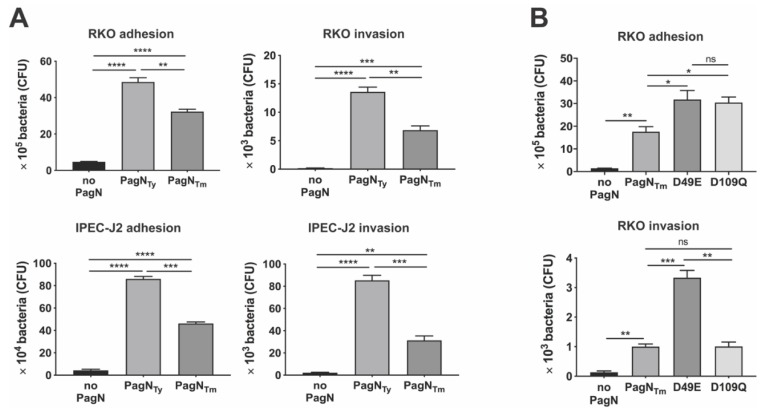
Binding and uptake of *E. coli* AAEC189 to intestinal epithelial cells mediated by PagN expression. (**A**) Adherence and invasion of *E. coli* AAEC189 with empty vector pRS1 (no pagN), pDMS1973 or pDMS1974 to express PagN_Ty_ or PagN_Tm_ respectively, to human (RKO) and porcine (IPEC-J2) was analyzed by incubating 3 × 10^7^ CFU/mL bacteria at a MOI of 100 for 60 min for adherence and another 90 min for invasion assays. Bacteria expressing PagN_Ty_ bound and invaded significantly better than the bacteria expressing PagN_Tm_ (*p* < 0.01–0.001). (**B**) Adherence of *E. coli* AAEC189 expressing PagN_Tm_ to RKO cells was significantly enhanced (*p* < 0.05) when PagN_Tm_ was mutated from aspartate to glutamate (at site 49, D49E) with pDMS2081 or to glutamine (at site 109, D109Q) with pDMS2082, whereas invasion of epithelial cells was affected only by the D49E substitution (*p* < 0.001), but not by the D109Q substitution. Data represent one of three separate and reproducible experiments each with triplicate data expressed as mean ± SEM (ns is for not significant, * *p* < 0.05, ** *p* < 0.01, *** *p* < 0.001 and **** *p* < 0.0001).

**Figure 4 microorganisms-08-00489-f004:**
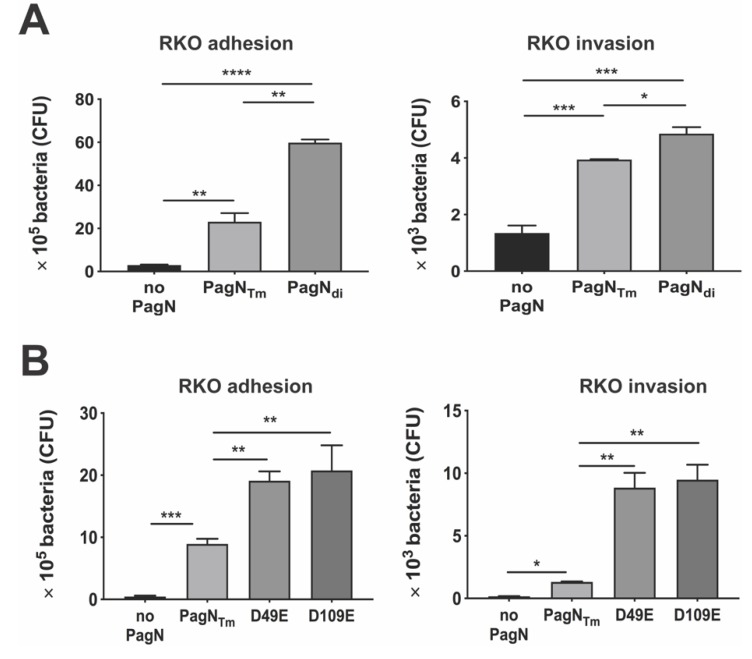
Binding and uptake of *E. coli* expressing PagN from non-*enterica* allelic variants. (**A**) Adhesion and invasion of *E. coli* AAEC189 expressing *pagN* allele from *S. diarizonae* PagN (PagN_di_) and *S.* Typhimurium PagN (PagN_Tm_) to RKO cells showed the bacteria expressing PagN_di_ binds significantly better than PagN_Tm_ (*p* < 0.01) and was slightly more invasive (*p* < 0.05). (**B**) A substitution of aspartate to glutamate at site 109 in *S.* Typhimurium PagN (D109E) increased both binding and invasion of *E. coli* into RKO cells as compared to the bacteria expressing PagN_Tm_ (*p* < 0.01). Data represent one of three separate and reproducible experiments, each with triplicate data expressed as mean ± SEM (* *p* < 0.05, ** *p* < 0.01, *** *p* < 0.001 and **** *p* < 0.0001).

**Figure 5 microorganisms-08-00489-f005:**
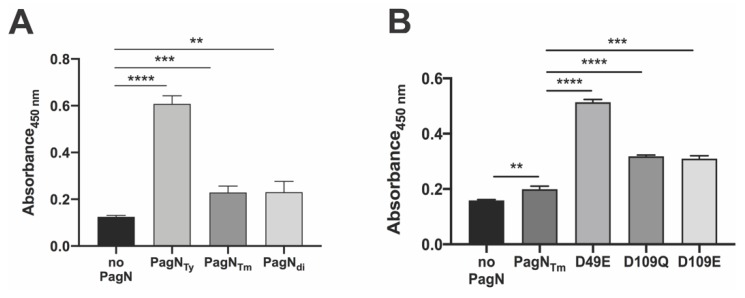
Binding of PagN allelic variants to murine laminin. (**A**) The PagN_Ty_-expressing bacteria bound best to laminin. *E. coli* AAEC189 expressing PagN_Tm_, PagN_Ty_, PagN_di_ or containing empty vector (no *pagN*) were incubated with microtiter wells coated with murine laminin. Bacterial binding was detected by an ELISA-method with anti-PagN antiserum. (**B**) All substitution mutants bound better to laminin than bacteria expressing PagN_Tm_, with the mutant expressing glutamic acid at position 49 of PagN_Tm_ (D49E) binding best (*p* < 0.0001). Data represent one of three separate and reproducible experiments, each with triplicate data expressed as mean ± SEM (** *p* < 0.01, *** *p* < 0.001 and **** *p* < 0.0001).

**Table 1 microorganisms-08-00489-t001:** Strains and plasmids.

Strain	Genus, Species, Serovar and/or Relevant Genotype	Source
AAEC189	*E. coli* MM294 ∆*lac recA endA* ∆*fim* (r_K−_ m_K+_)	[32]
BL21(DE3)	*E. coli* str. B F^−^ *ompT gal dcm lon hsdS_B_*(*r_B_*^−^*m_B_*^−^) λ(DE3 [*lacI lacUV5*-*T7p07 ind1 sam7 nin5*]) [*malB*^+^]_K-12_(λ^S^)	Novagen
JKE201	*E. coli* MG1655 RP4-2-Tc::[ΔMu1::*Δaac(3)IV*::*lacI^q^-ΔaphA-Δnic35*-ΔMu2::*zeo*] Δ*dapA::(erm-pir)* Δ*recA* Δ*mcrA* Δ(*mrr-hsdRMS-mcrBC*))	[29,33]
SL1344	*S. enterica* subsp. *enterica* serovar Typhimurium	[34]
*S.* Typhi	*S. enterica* subsp. *enterica* serovar Typhi 9,12,[Vi]:d:[Z66]	*Salmonella* Reference collection C (SARC) no2
*S. diarizonae*	*S. enterica* subsp. *diarizonae* 50:k:z	SARC no7
DMS1507	*S. enterica* subsp. *enterica* serovar Typhimurium LB5010 *sipB::aphA-3*	[27,28]
DMS1949	SL1344 *sipB* Δ*pagN*	This study
**Plasmid**	Description	Source
pMG81	Expression plasmid with *tetR* and *tetA* promoter upstream *envZ*, Ap^R^	Mark Goulian
pRS1	pMG81∆*envZ*	This study
pDMS1973	*S.* Typhi *pagN* in pRS1	This study
pDMS1974	SL1344 *pagN* in pRS1	This study
pDMS2062	*S. diarizonae pagN* in pRS1	This study
pDMS2081	SL1344 *pagN*_D49E_ in pRS1	This study
pDMS2082	SL1344 *pagN_D109Q_* in pRS1	This study
pDMS2083	SL1344 *pagN_D109E_* in pRS1	This study

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
