# Peer review of "The Not so Good, the Bad and the Ugly: Differential Bacterial Adhesion and Invasion Mediated by *Salmonella* PagN Allelic Variants"

_microorganisms, 2020, doi:10.3390/microorganisms8040489_

Round 1

Reviewer 1 Report

The manuscript by Yanping Wu et al. investigated the impact of allelic variation of Salmonella adhesin/invasin PagN in bacterial interaction with host receptor molecules. The article reads well. However, all experiments we done using transformed E. coli expressing different PagN variants. The authors did not try to generate the PagN variants in Salmonella strains. I would introduce all the PagN variants in both Salmonella Typhi and Typhimurium strains and check the adhesion/invasion phenotype. Although E. coli expressing different PagN variants show changes in adhesion/invasion capacity, since Salmonella Typhi and Typhimurium infection mount different responses in the host, it is important to know whether the adhesion/invasion phenotypes observed in E. coli are true/relevant for the Salmonella strains as well. Replication of E. coli(PagN) adhesion/invasion phenotypes in Salmonella Typhi/Typhimurium might be significant in understanding the contribution of PagN variants in Salmonella pathogenesis. 

Minor- quality of all images needs improvement. 

Author Response

The manuscript by Yanping Wu et al. investigated the impact of allelic variation of Salmonella adhesin/invasin PagN in bacterial interaction with host receptor molecules. The article reads well.

We appreciate Reviewer 1 for his positive view on our study.

The authors did not try to generate the PagN variants in Salmonella strains. I would introduce all the PagN variants in both Salmonella Typhi and Typhimurium strains and check the adhesion/invasion phenotype.

Although we undertook this experiment (e.g. express PagN of S. Typhi or S. Typhimurium in pagN mutants) we did not obtain significant differences in binding or invasion between the PagN alleles, likely due to the large number of other Salmonella adhesins and/or invasins that are expressed under the in vitro conditions used (PLoS One 2012, 7:e38596; Sci Rep. 2017, 7:10326). Since PagN-mediated effects on adhesion and invasion was more significant when studied in the context of a laboratory strain of E. coli (manuscript reference 12), it was clear that the individual effects of PagN alleles would be best detected in a bacterial context that is free of other Salmonella adhesins. Accordingly, and as done in our previous work on Salmonella fimbrial adhesin alleles (reference 25), we used a laboratory strain of E. coli which lacks type 1 fimbriae together with a controlled expression system for the PagN alleles.

The results of our experiments with Salmonella and our interpretation of the data were added to the Results and Discussion section of the manuscript (lines 217-223). Salmonella mutant construction is now described in Materials and Methods (lines 111-114) with the corresponding addition to Table S1.

Although E. coli expressing different PagN variants show changes in adhesion/invasion capacity, since Salmonella Typhi and Typhimurium infection mount different responses in the host, it is important to know whether the adhesion/invasion phenotypes observed in E. coli are true/relevant for the Salmonella strains as well.

We fully agree that both Salmonella serovars induce different responses in hosts. However, the aim of this study was not to investigate host responses due to PagN-mediated adhesion/invasion during a Salmonella infection but to examine the differential effect of PagN allelic variants on the known property of this adhesin/invasin. Here, we illustrated this effect on the adhesion/invasion of human intestinal epithelial cells with three relevant serovars responsible for different forms of diseases in humans. Whether “the adhesion/invasion phenotypes observed in E. coli are true/relevant for the Salmonella strains” is an important question that was already published individually for Salmonella Typhi and Typhimurium (reference 12 and 23), as presented in the Introduction of the manuscript (lines 62-64 and 72-74). Decreased adhesion and invasion with various cell lines were described for PagN deletion mutants in either serovars, albeit different results were obtained with different strains of S. Typhimurium (lines 74-76). In addition, reference 23 found that orally challenged iron-overloaded Swiss Albino mice were more susceptible to a wild type strain of S. Typhi than its isogenic pagN mutant, whereas no differences were observed between the wild type and pagN mutant strains of S. Typhimurium (lines 81 to 84).

Replication of E. coli(PagN) adhesion/invasion phenotypes in Salmonella Typhi/Typhimurium might be significant in understanding the contribution of PagN variants in Salmonella pathogenesis.

Unfortunately, as mentioned above and in the Introduction of the manuscript (e.g. lines 85-88), variable panoplies of other expressed adhesins/invasins might have had a dominant effect on the studied phenotypes that obscured the comparison of wild type strain and pagN mutant phenotypes in the literature (lines 74-76) and in our assays using Salmonella strains with the different PagN alleles.  The use of E. coli and a controlled expression system allowed us to compare bacterial adhesion/invasion mediated by diverse PagN alleles in a neutral environment devoid of confounding Salmonella surface molecules differentially expressed in vitro and in vivo. To really understand the role of PagN of S. Typhi in human pathogenesis would require unrealistic experiments in humans, and whereas the mouse model used in reference 23 is informative, it might not necessarily reflect what happens in humans. We understand the limitations of our study, but nevertheless strongly believe that it adds significantly to a better understanding of PagN and provides another example of the contribution of allelic virulence factor variants in bacterial pathogenesis.

Minor- quality of all images needs improvement.

Better quality of the images will be uploaded/sent as TIF files if the manuscript is accepted for publication.

Reviewer 2 Report

The manuscript by Wu et al investigated the role of PagN proteins in the adhesion of Salmonella to epithelial cells. It has been established that PagN, a homolog of Tia/Hek adhesins of E. coli, mediates Salmonella adhesion to host cells likely via glycoproteins. By examining the sequence conservation of PagN in different Salmonella strains, the authors identified a number of amino acid variants located in the flexible loops of PagN. Interestingly, the S. typhi version of PagN is more effective in both adhesion and invasion, as well as the S. diarizonae version of PagN. Mutagenesis experiments could further confirm the critical contribution of D49 and D109 amino acids in the loop 1 and loop2, respectively. Furthermore, the authors could show that PagN of S. typhi also binds to the host laminin protein, and that loop1 (D49) contributes to this interaction in a heparin-independent manner. Altogether, this manuscript improved the molecular understanding of the PagN protein, a less-understood adhesin in Salmonella.

One minor suggestion is to include the sequence alignment of loop 1 and 2 as a main figure, as this information is key to the manuscript.  

Author Response

The manuscript by Wu et al investigated the role of PagN proteins in the adhesion of Salmonella to epithelial cells. It has been established that PagN, a homolog of Tia/Hek adhesins of E. coli, mediates Salmonella adhesion to host cells likely via glycoproteins. By examining the sequence conservation of PagN in different Salmonella strains, the authors identified a number of amino acid variants located in the flexible loops of PagN. Interestingly, the S. typhi version of PagN is more effective in both adhesion and invasion, as well as the S. diarizonae version of PagN. Mutagenesis experiments could further confirm the critical contribution of D49 and D109 amino acids in the loop 1 and loop2, respectively. Furthermore, the authors could show that PagN of S. typhi also binds to the host laminin protein, and that loop1 (D49) contributes to this interaction in a heparin-independent manner. Altogether, this manuscript improved the molecular understanding of the PagN protein, a less understood adhesin in Salmonella.

We thank Reviewer 2 for his appreciation of our study on PagN allelic variants.

The author had One minor suggestion, namely “to include the sequence alignment of loop 1 and 2 as a main figure, as this information is key to the manuscript.”

Done. Old Fig. S1 is now included as panel B of Fig. 1 (and old Fig. 1 is now panel A of this figure).

We thank you both for your constructive comments and suggestions.

Round 2
